# Persistent BK Polyomavirus Viruria Is Associated with Accumulation of VP1 Mutations and Neutralization Escape

**DOI:** 10.3390/v12080824

**Published:** 2020-07-29

**Authors:** Dorian McIlroy, Mario Hönemann, Ngoc-Khanh Nguyen, Paul Barbier, Cécile Peltier, Audrey Rodallec, Franck Halary, Emilie Przyrowski, Uwe Liebert, Maryvonne Hourmant, Céline Bressollette-Bodin

**Affiliations:** 1Centre de Recherche en Transplantation et Immunoologie (CRTI), UMR 1064, INSERM, Université de Nantes, 44093 Nantes, France; ngoc-khanh.nguyen@univ-nantes.fr (N.-K.N.); paul.barbier@univ-nantes.fr (P.B.); cecile.peltier@univ-nantes.fr (C.P.); franck.halary@univ-nantes.fr (F.H.); celine.bressollette@univ-nantes.fr (C.B.-B.); 2Institut de Transplantation Urologie-Néphrologie (ITUN), CHU Nantes, 44093 Nantes, France; maryvonne.hourmant@chu-nantes.fr; 3Faculté des Sciences et des Techniques, Université de Nantes, 44322 Nantes, France; 4Institut für Virologie, Universität Leipzig, 04103 Leipzig, Germany; Mario.Hoenemann@medizin.uni-leipzig.de (M.H.); liebert@medizin.uni-leipzig.de (U.L.); 5Service de Virologie, CHU Nantes, 44093 Nantes, France; Audrey.RODALLEC@chu-nantes.fr (A.R.); EmPrzyrowski@chu-angers.fr (E.P.); 6Service de Néphrologie et Immunologie Clinique, CHU Nantes, 44093 Nantes, France; 7Faculté de Médecine, Université de Nantes, 44093 Nantes, France

**Keywords:** polyomavirus, transplantation, immunity, neutralization

## Abstract

To investigate the relationship between neutralization escape and persistent high-level BK polyomavirus replication after kidney transplant (KTx), VP1 sequences were determined by Sanger and next-generation sequencing in longitudinal samples from KTx recipients with persistent high-level viruria (non-controllers) compared to patients who suppressed viruria (controllers). The infectivity and neutralization resistance of representative VP1 mutants were investigated using pseudotype viruses. In all patients, the virus population was initially dominated by wild-type VP1 sequences, then non-synonymous VP1 mutations accumulated over time in non-controllers. BC-loop mutations resulted in reduced infectivity in 293TT cells and conferred neutralization escape from cognate serum in five out of six non-controller patients studied. When taken as a group, non-controller sera were not more susceptible to neutralization escape than controller sera, so serological profiling cannot predict subsequent control of virus replication. However, at an individual level, in three non-controller patients the VP1 variants that emerged exploited specific “holes” in the patient’s humoral response. Persistent high-level BK polyomavirus replication in KTx recipients is therefore associated with the accumulation of VP1 mutations that can confer resistance to neutralization, implying that future BKPyV therapies involving IVIG or monoclonal antibodies may be more effective when used as preventive or pre-emptive, rather than curative, strategies.

## 1. Introduction

The BK polyomavirus PyV (BKPyV) is a typical opportunistic pathogen. Following asymptomatic primary infection during childhood, it establishes a latent infection in the kidney, which appears to persist throughout life. Approximately 7% of healthy adults excrete BKPyV in the urine [1], and this proportion increases during natural, acquired [2], or iatrogenic [3] immunosuppression. Its pathogenic potential is manifested in kidney transplant (KTx) recipients, in whom uncontrolled BKPyV replication can result in polyomavirus nephropathy (PyVAN) and graft loss or dysfunction. PyVAN can only be diagnosed definitively by histology, but it is correlated with viremia greater than 10^4^ genome copies/mL [4], and high-level viremia is generally classified as presumptive PyVAN [5]. Recent results indicate that at least 90% of BKPyV genomes in plasma are DNAse sensitive, that is, represent free DNA and not infectious virions [6], which explains why BKPyV replication remains restricted to the graft and does not spread to the native kidneys. There is currently no approved antiviral therapy with clinical efficacy against BKPyV, so presumptive or biopsy-confirmed PyVAN is managed by modulation of immunosuppressive therapy, which allows host immune responses to clear the virus [7]. The virological response rate to this intervention appears to vary between centres, with recent publications reporting clearance of viremia in response to modulation of immunosuppression in proportions varying from 30% [8,9] up to more than 75% [10] of PyVAN patients. In the single-centre study with the longest follow-up and the largest cohort, clearance of viremia was attained by 44 (92%) of 48 patients with presumptive PyVAN following a standard three-stage protocol for reduction of immunosuppressive therapy [11]. However, the interquartile range for time to viremia clearance was 65–414 days, meaning that at least 25% of these PyVAN patients had viremia that persisted for more than 1 year, despite modulation of immunosuppressive therapy. Similarly, the Banff working group on PyVAN, analyzing data from nine transplant centres in Europe and North America, found that PyVAN persisted for more than 24 months in 39 of 149 (26%) patients [12].

The importance of the neutralizing antibody response in controlling BKPyV replication in KTx recipients has been highlighted by the observation that higher neutralizing titres at the time of graft are correlated with lower risk of BKPyV reactivation in the 12 months following KTx [13]. Surprisingly, the strong humoral response that develops in KTx recipients after BKPyV reactivation [13,14] does not correlate with subsequent viral clearance, and several studies have implicated CTL rather than antibodies as playing the dominant role [15,16,17] in the resolution of BKPyV infection after modulation of immunosuppressive therapy. In HIV [18,19] and HCV [20,21] infection, persistent viremia in the face of a robust humoral response is the result of the selection of escape mutations in viral glycoproteins, leading to an evolutionary arms race between the virus and the host neutralizing response, in which host responses continually struggle to keep up with a rapidly evolving virus population. Because polyomaviruses are generally considered to be slowly evolving DNA viruses, a similar process was not thought to be plausible during BKPyV infection. Nevertheless, mutations in the major BKPyV capsid protein, VP1, have been observed in patients with PyVAN [22,23], and appear to accumulate in the typing region of the VP1 gene [24,25], which distinguishes the four different BKPyV genotypes, and codes for the BC-loop on the surface of the virus capsid. Results published in 2018 provided an elegant explanation of how these mutations arise and the first description of their functional impact [26]. In two KTx patients with PyVAN caused by genotype IV BKPyV, Peretti et al. showed firstly, that mutations in VP1 carried the signature of cytosine deamination on the antisense strand by the apolipoprotein B mRNA editing, catalytic polypeptide-like (APOBEC) 3B enzyme; secondly, that these mutations led to resistance against neutralization; and thirdly, that APOBEC3 enzymes are expressed in renal biopsies from KTx recipients. However, the patients included in this study were both characterized by a rather uncommon clinical presentation of clear cell renal carcinoma concomitant with, or subsequent to PyVAN. In addition, both patients cleared their BKPyV viremia, despite the presence of apparent neutralization escape mutations. Therefore, it is not entirely clear whether the results of the study by Peretti et al. can be generalized to other BKPyV genotypes and to more frequent clinical situations, and the relationship between the presence of VP1 mutations and viral clearance remains an open question. In order to address these points, we first investigated the presence of mutations in the VP1 typing region in KTx recipients with persistent or transient high-level BKPyV viruria, then analyzed the functional impact of BC-loop mutations in genotype I and genotype IV VP1 proteins on infectivity and resistance to neutralization.

## 2. Materials and Methods 

### 2.1. Patients and Clinical Samples

Patients in the present study were included retrospectively in Nantes and in Leipzig, based on duration of viral load and BKPyV genotype recorded in the hospital laboratory databases. Patients in Nantes were transplanted between 2011 and 2014 and had previously been included in a prospective observational study, approved by the local ethics committee (approval date: 8 November 2011) and declared to the French Commission Nationale de l’Informatique et des Libertés (CNIL, n°1600141). Patients in Leipzig were transplanted between 2009 and 2012. Retrospective analysis was approved by the local ethics committee (Leipzig University ethical review committee ref.-no. 300/16-ek, approval date: 26 September 2016). All patients in Nantes and Leipzig gave informed consent authorizing the use of archived urine and blood samples for research protocols. Anonymised clinical and biological data for these patients were extracted from the hospital databases. Patients 3.16, 3.17 and 3.18 were recruited in Leipzig; all other patients were recruited in Nantes.

In order to define two clearly distinct patient groups, KTx recipients in Nantes and Leipzig with peak viruria >7 log10 copies/mL were stratified into controllers, who showed a drop in viruria of greater than 2 log10 copies/mL at 6 months following peak viruria, and a reduction of greater than 3 log10 copies at 12 months following peak viruria (*n* = 12, Figure 1A), and non-controllers who had a drop in viruria less than 2 log10 copies/mL over 12 months (*n* = 12, Figure 1B). Patients with high-level viruria that lasted more than 6 months, but less than 12 months, were considered “intermediate”, and not included. Presence of viremia was not used as an inclusion criterion as preliminary investigations had shown that VP1 mutations could be detected in patients with persistent viruria in the absence of viremia.

### 2.2. VP1 Sequence Analysis

For Sanger sequencing, the typing region was amplified from urine and whole-blood extracted DNA using primers described in Takasaka et al. [27], and the full-length VP1 gene was amplified using primers described in Sharma et al. [28]. After Sanger sequencing of PCR products, VP1 sequences were analyzed using SeqScape™ software v3.0 (ThermoFisher Scientific). BKPyV genotypes were identified, and nucleotide polymorphisms were accepted only if present on both sense and antisense sequences. Sequences containing at least one non-synonymous BC-loop mutation were selected, and in the case of multiple sequences from the same patient, the sequence containing the most mutations was retained, so that each observed mutation represented an independent event. The prevalence of each mutation was calculated as the number of times each mutation was observed divided by the number of sequences.

To illustrate the positions of these mutated amino acids on the virus capsid, BKPyV VP1 pentamers were visualized using Pymol 1.8.4.0 under a Ubuntu Linux operating system. The 4MJ0 crystal structure of wild-type genotype I VP pentamer in complex with GD3 oligosaccharide was loaded, then rotated in order to present the sialic-acid binding pocket with its associated oligosaccharide ligand at the centre of the figure. The surfaces of the central (C), clockwise (CW), and counter-clockwise (CCW) VP1 subunits were rendered using the “show surface” command, and coloured light green, light blue, and light grey, respectively. The GD3 oligosaccharide was shown in a yellow stick representation. Since no three-dimensional structure of genotype IV BKPyV VP1 has been described, for visualization purposes, the genotype IV-specific amino acids were introduced using the Wizard/Mutagenesis function. Specifically, the S71T, N74T, D75A, and S77D mutations were introduced on the CW VP1 subunit, and the E61N, F66Y, K69R, and E82D mutations were introduced on the C subunit. Finally, amino acid residues that were found to be mutated in patients were coloured on a scale from light orange to red, according to the prevalence of mutations found at each site. Specifically, positions 59, 60, 61, 62, 68, 69, and 82 were coloured on the C subunit, positions 72, 73, 75, 66, and 170 were coloured on the CW subunit, and positions 138 and 139 were coloured on the CCW subunit. 

For next-generation sequencing (NGS) of the typing region, eight different unique and base-balanced 8bp barcodes were added to the 5’ end of each forward and reverse primer, respectively. This provided a 16 primer set from which each forward and reverse primer were used in combination to create a dual barcode sample index. Barcoded VP1 sequences were amplified in a reaction volume of 25 µL containing 5 µL extracted viral DNA, 0.5 U Platinum SuperFi DNA Polymerase (Thermo Fisher, Courtaboeuf, France), 200 µM each dNTP, 0.5 µM primers, and 1x PCR buffer. After initial denaturation at 98 °C for 3 min, amplification was performed for 35 cycles on a LifePro Thermal Cycler (Dutscher, Issy-les-Moulineaux, France). Cycling conditions were 98 °C for 30 s, 54 °C for 30 s, and 72 °C for 30 s, with final extension at 72 °C for 5 min. PCR products were purified using columns (NucleoSpin Gel and PCR Clean-up, Macherey-Nagel, Hoerdt, France) and eluted into 5 mM Tris/HCl, pH 8.5 buffer. Amplicon lengths and concentrations were measured with a LabChip GX (PerkinElmer, Villebon-sur-Yvette, France) using a high sensitivity 1K chip. PCR products were then pooled in equimolar proportions then sent for DNA library preparation and Illumina sequencing at GENEWIZ, Inc. (South Plainfield, NJ, USA).

Illumina PE raw reads were quality trimmed, checked for N content and length filtered using Cutadapt version 2.3 [29]. Reads were then 5’ trimmed and assigned to each patient’s sample using Flexbar [30] version 3.3.0, and Illumina reads one and two were merged with NGmerge [31]. Mutations and their associated frequencies were predicted by running the breseq pipeline version 0.33.0 [32] with targeted-sequencing and polymorphism-mode options with default parameters using the Dunlop BKPyV genome (Accession Number NC_001538.1) as reference. After quality trimming and alignment 1263–8163 (mean 3449 ± 1543), merged sequences were analyzed per sample. NGS data has been deposited as a Targeted Locus Study in the following NCBI Bioproject PRJNA633999: Intra patient evolution of Human polyomavirus 1 (BKPyV).

### 2.3. Cell Culture

Vero cells and HEK 293TT cells, purchased from the National Cancer Institute’s Developmental Therapeutics Program, were maintained in DMEM High Glucose (Thermo-Fisher) containing 10% FBS (Dutscher), 100 U/mL penicillin, 100 µg/mL streptomycin (Thermo-Fisher), 1x Glutamax (Thermo-Fisher), and 250 µg/mL Hygromycin (Sigma-Aldrich, Lyon, France). RS cells (Evercyte, Vienna, Austria), which are immortalized human renal tubular epithelial cells, were cultured in Proxup-3 medium (Evercyte, Vienna, Austria) supplemented with 100 U/mL penicillin and 100 µg/mL streptomycin (Thermo-Fisher) in tissue-culture plasticware coated with 50 µg/mL collagen I (Thermo-Fisher). 293TT and RS cells were grown at 37 °C with 5% CO2 in a humidified incubator, and passaged at confluence by trypsinization for 10 min with 1x Trypsin-EDTA in PBS (Thermo-Fisher). Cultures were routinely tested for mycoplasma contamination by PCR [33] and were consistently negative.

### 2.4. Plasmids

The VP2 and VP3 expression plasmids ph2b and ph3b (#32109 and #32110) were obtained from Addgene (Cambridge, MA, USA). VP1 expression plasmids for genotypes Ia, Ib2 and IVc2 were kindly provided by Dr Christopher Buck, National Cancer Institute (NCI), Bethesda, MD. The plasmid pEGFP-N1 (Clontech) was used as the reporter gene. Mutations were introduced into the relevant VP1 plasmids by site-directed mutagenesis using the Q5 Site-directed mutagenesis kit (New England Biolabs, Evry, France). Primer pairs used for mutagenesis were selected using the NEBasechanger tool. After each mutagenesis reaction, miniprep DNA from four colonies was screened by Sanger sequencing (Eurofins Genomics, Ebersberg, Germany) with the EF1a-F primer. The full-length VP1 sequence of clones incorporating the desired mutations was confirmed by sequencing with the WPRE-R primer. The BKPyV-MM genome cloned into the pBR322 vector (pBKV 35-1, ATCC 45026) was obtained from the ATCC, and VP1 mutations were introduced using the Q5 Site-directed mutagenesis kit (New England Biolabs). 

### 2.5. Pseudotype BKPyV Production

Pseudotype BKPyV (PSV) particles were prepared following a slightly modified version of protocols developed by the Buck lab [34,35]. Briefly, 293TT cells were seeded at 1 × 10^7^ cells in a 75 cm^2^ flask in DMEM 10% FBS without antibiotics, then co-transfected using Lipofectamine 2000 reagent (Thermo-Fisher) according to manufacturer’s instructions. A total of 36 µg plasmid DNA comprising 16 µg VP1 plasmid, 4µg ph2b, 8 µg ph3b, and 8 µg pEGFP-N1 was mixed with 1.5 mL of Opti-MEM I (Thermo-Fisher). Eighty-five microlitres of Lipofectamine 2000 was diluted in 1.5 mL of Opti-MEM I and incubated for 5 min at room temperature, then mixed with the diluted plasmid DNA. After 20 min at room temperature, DNA-Lipofectamine complexes were added to each flask containing pre-plated 293TT cells.

Producer cells were harvested by trypsinisation 48 h after transfection. The pellet was washed once in cold PBS then resuspended in 800 µL hypotonic lysis buffer consisting of 25 mM Sodium Citrate pH 6.0, 1 mM CaCl_2_, 1 mM MgCl_2_, and 5mM KCl. Cells were sonicated in a Bioruptor Plus device (Diagenode, Seraing, Belgium) for 10 min at 4 °C with 5 cycles of 1 min. ON/1 min OFF. Type V neuraminidase (Sigma-Aldrich) was added to a final concentration of 1 U/mL and the extract was incubated for 30 min. at 37 °C. The pH was neutralized by adding 100 µL of 1M HEPES buffer pH 7.4, then 1 µL (250 U) Pierce Nuclease (Thermo-Fisher) was added followed by 2 h incubation at 37 °C. The lysate was clarified by centrifuging twice at 5000× *g* for 5 min at 4 °C, then layered onto an Optiprep 27%/33%/39% step gradient prepared in DPBS/0.8M NaCl. Gradients were centrifuged overnight at 175,000× *g* at 4 °C in an SwTi55 rotor. Tubes were punctured with a 25G syringe needle, and 10 fractions of each gradient were collected into 1.5 mL microcentrifuge tubes. Eight microlitres of each fraction was removed for qPCR, and PBS 5% BSA was then added to each fraction to give a final concentration of 0.1% BSA as a stabilizing agent before tubes were transferred to −80 °C. The two or three peak fractions from each pseudo-virus preparation were pooled and aliquoted for use in infectivity and neutralisation assays.

For quantification of reporter plasmid, 8 µL of each fraction was mixed with 2 µL of proteinase K buffer containing 100 mM Tris-HCl pH 7.5 (Thermo-Fisher), 100 mM DTT (Sigma), 25 mM EDTA (Sigma), 1% SDS (Sigma) and 200 µg/mL proteinase K (Qiagen, Courtaboeuf, France). This solution was incubated at 50 °C for 60 min, then heated to 95 °C for 10 min and diluted 80-fold in water. One microlitre of this diluted solution was used for qPCR using Applied Biosystems 2× Sybr Mix (Applied Biosystems, Villebon-sur-Yvette, France). Primers were CMV-F 5′-CGC AAA TGG GCG GTA GGC GTG-3′ and pEGFP-N1-R 5′-GTC CAG CTC GAC CAG GAT G-3′. Thermal cycling was initiated with a first denaturation step at 95 °C for 10 min to activate the polymerase, followed by 35 cycles of 95 °C for 15 s and 55 °C for 40 s. Standard curves were constructed using serial dilutions from 10^2^ to 10^7^ copies of the pEGFP-N1 plasmid.

### 2.6. Infectivity Assay

Cells were seeded at 10^4^ cells/well in 96-well Falcon plates (BD Falcon, Le Pont de Claix, France) then left to adhere at 37 °C for at least 1 h. Serial dilutions of each type of BK pseudo-virus were prepared in DMEM 10% FBS + antibiotics, then added to plated 293TT or RS cells so as to inoculate from 1 × 10^7^ to 8 × 10^4^ pEGFP-N1 copies per well of each VP1 variant in quadruplicate. Plates were incubated in a humidified 5% CO_2_ incubator at 37 °C for 72 h (293TT cells), or 96 h (RS cells) after which, cells were washed once in PBS 0.5 mM CaCl_2_, 0.5 mM MgCl_2_, then fixed and stained in PBS 0.5 mM CaCl_2_, 0.5 mM MgCl_2_, 1% paraformaldehyde, and 10 µg/mL Hoechst 33342. The number and percentage of GFP^+^ cells was quantified using a Cellomics ArrayScan VTI HCS Reader (Thermo Scientific, Courtaboeuf, France). Twelve to twenty-five fields containing 5000–20,000 cells were acquired for each well using HCS Studio Cellomics Scan Version 6.5.0 software. The log10 percentage of GFP^+^ cells was plotted against the log10 inoculated dose (Appendix A), and the linear part of the curve was used to calculate the dose required to infect 1% of the plated cells, and hence the ratio of infectious particles to pEGFP-N1 copies in the pseudovirus stock. Control experiments (Appendix A) showed that the infectious particle/pEGFP-N1 ratio was reproducible across different quadruplicates set up in the same experiment, and that independent PSV preparations with wild-type VP1 had identical infectious particle/pEGFP-N1 ratios, when measured in the same experiment. To test the dependence of PSV entry on sialic acid, RS or 293TT cells were plated at 10^4^ cells/well in 96-well plates, cultured overnight, then parallel series of wells were either left untreated, or incubated with 0.5 U/mL Type V neuraminidase (Sigma) in DMEM supplemented with 0.1% BSA and 25 mM HEPES pH 7.4 for 1 h at 37 °C. Cells were washed three times in PBS 5% FCS, then incubated with PSV at 5 × 10^7^ copies EGFP per well in culture medium for 1 h at 37 °C. Medium was removed, cells were washed three times in PBS 5% FCS, then culture medium was added and cells were incubated for 48–72 h (293TT cells) or 72–96 h (RS cells) before quantification of GFP+ cells as described above. For each PSV, the percentage of GFP+ cells observed in the presence and absence of neuraminidase was used to calculate the percentage inhibition of infectious entry by neuraminidase treatment.

To test infectivity of the whole virus, 50 µg of the pBKV 35-1 plasmid carrying wild-type or mutant VP1 was digested with BamHI and PvuI (New England Biolabs) in CutSmart buffer. Cleavage products were purified on DNA Nucleospin columns (Macherey-Nagel), quantified using a Nanodrop spectrophotometer, then 20 µg of cut DNA were recircularized by ligation with T4 ligase overnight at 16 °C in a reaction volume of 1.2 mL to favour self-ligation. Ligation products were precipitated with isopropanol, then redissolved in sterile 10 mM Tris pH 8, 1 mM EDTA. For transfection, Vero cells were seeded into a 96-well plate at 1 × 10^4^ cells per well, then transfected the next morning with 200 ng/well religated pBKV 35-1 using the Lipofectamine 2000 reagent (Thermo-Fisher). Twenty-four hours after transfection, medium was removed, cells were washed in PBS to remove excess plasmid, and fresh medium was added to wells. At day 7 and 14 post transfection, cells were washed in PBS, then fixed in 2% PFA in PBS for 20 min at room temperature (RT), washed three times in PBS, then permeabilized with 1% Triton X-100 in PBS for 15 min at 37 °C. After three washes in PBS, cells were blocked with 0.5% BSA in PBS for 1 h at RT, then incubated with the mouse monoclonal anti-SV40 TAg (PAb416 clone, Abcam, Cambridge, UK) diluted 1/200 in PBS 0.1% BSA for 1 h at RT. After three washes in PBS 0.05% Tween-20, cells were incubated with Alexa-488 conjugated goat anti-mouse IgG (Thermo Scientific) diluted 1/200 in PBS 0.1% BSA for 1 h at RT before three final washes in PBS 0.05% Tween-20, then counterstaining with PBS 0.5 mM CaCl_2_, 0.5 mM MgCl_2_, 1% paraformaldehyde and 10 µg/mL Hoechst 33342. The percentage of cells positive for TAg staining was quantified using a Cellomics ArrayScan VTI HCS Reader (Thermo Scientific) as described above.

### 2.7. Neutralization Assay

Cells were seeded as for an infectivity assay, and during adhesion, pseudovirus stocks were diluted in DMEM supplemented with 0.1% BSA and 25 mM HEPES pH 7.4, and distributed in a 96-well U-bottom plate. Decomplemented patient serum was added to the first well, then serially diluted in the plate before incubation for 60 min at 4 °C. Pseudotype virus incubated with diluted serum was then transferred onto pre-plated cells. Plates were incubated in a humidified 5% CO2 incubator at 37 °C for 72 h (293TT cells), or 96 h (RS cells), then fixed and analyzed as described above. A sufficient number of fields were acquired in order to count a minimum of 200 GFP^+^ cells in control wells incubated in the absence of serum. Percentages of positive cells in test wells were normalized with respect to these control wells, and log10 neutralizing titers were calculated using Prism 5™ software. To compare the effects of different VP1 mutations on neutralization escape, neutralizing titres in each serum were normalized by subtracting the log10 neutralizing titre against wild-type VP1 from the log10 neutralizing titre observed against the variant in the same serum sample.

### 2.8. Statistical Analysis

Between patient groups, categorical variables were compared by Fisher’s exact test and continuous variables were compared by unpaired t-test. Infectious particle/pEGFP-N1 ratios were compared by one-way ANOVA followed by Dunnett’s post-hoc test, using the wild-type gIb2-E^73^E^82^, gIVc2, or gIVc2-S^61^ PSV as control for all comparisons. Fisher’s exact test was performed using the http://statpages.info/ctab2x2.html webpage, and all other tests were performed using Graphpad Prism 5.

## 3. Results

### 3.1. VP1 Mutations Accumulate in KTx Recipients with Persistent BKPyV Viruria

In order to examine the relationship between the intensity and duration of BKPyV replication, the presence of non-synonymous *VP1* mutations, and the neutralizing antibody response, we studied BKPyV *VP1* sequences obtained from the urine of KTx recipients in Nantes and Leipzig with peak viruria >7 log10 copies/mL. Patients who showed a rapid and sustained reduction in viruria (>3 log10 drop in viruria at 12 months after peak) were defined as the controller group (*n* = 12, Figure 1A), and those who had a drop in viruria less than 2 log10 copies/mL over the same period were defined as non-controllers (*n* = 12, Figure 1B).

BKPyV controllers and non-controllers had comparable age, male–female sex ratio, and received similar induction and maintenance immunosuppressive therapies (Table 1). Concerning virological parameters, the delay from graft to first positive viruria, peak viruria, peak viremia, and the proportion of patients with detectable viremia and histologically confirmed PyVAN, were all similar in controllers and non-controllers, and BKPyV genotype I viruses predominated in both groups. The cold ischemia time (CIT) was significantly shorter in the non-controller group (8.7 ± 4.8 h compared to 17.9 ± 6.6 h, *p* < 0.001), and this difference remained significant after excluding three kidneys that came from living donors in the non-controller group. Furthermore, the mean number of HLA mismatches was higher in non-controllers than in controllers.

In controllers, the kinetics of viruria clearance could be modelled as a single-phase (*n* = 10) or two-phase (*n* = 2) exponential decay curve preceded in some cases by a plateau of up to 90 days, with a mean viruria half-life of 6.4 ± 3.2 days, corresponding to a reduction in urine viral load greater than 1 log10 per month during the decay phase. In contrast, viruria remained higher than 7 log10 copies/mL for more than 12 months in 10 of 12 non-controllers, and the mean reduction in viruria over that time was less than 0.5 log10. Changes in viremia mirrored changes in viruria. In controllers, viremia was durably suppressed (Figure 1C), whereas in non-controllers, viremia remained close to initial values: patients with high (>4 log10 copies/mL) peak viremia maintained plasma viral loads varying from 3 to 6 log10 copies/mL; patients with intermediate (>3 log10 copies/mL) peak viremia had durable viremia from 2.3 to 4 log10 copies per mL, while three non-controllers had low or undetectable viremia throughout, despite persistent high-level viruria (Figure 1D).

Sequential urine samples were subjected to PCR and Sanger sequencing, and the number of non-synonymous *VP1* mutations in the BC-loop (amino acids 59-84) was plotted against time following initial viruria > 7 log10 copies/mL (Figure 1E,F). The BC-loop sequence was identical to the reference wild-type sequence at the first time point analyzed in 11 out of 12 controllers and 10 out of 12 non-controllers, and sequences in the three remaining patients differed by only one amino acid. The virus population in both controllers and non-controllers was therefore dominated by wild-type virus at early time points. Subsequently, there was a clear correlation between the number BC-loop mutations and the duration of high-level viruria in non-controllers (*r*^2^ = 0.67, *p* < 0.0001). Comparing VP1 sequences in sequential samples from the same patient confirmed that BC-loop mutations accumulated over time in non-controllers (Figure 1F), but this was not observed in the rare controller patients with persistent low-level viruria (Figure 1E). Furthermore, the accumulation VP1 mutations over time was also observed in patients with undetectable or low viremia (blue symbols, Figure 1E). When possible, *VP1* sequences were analyzed from blood drawn on the same day as the urine sample (18 pairs of samples from 8 patients). In all but one case, the same *VP1* sequences were found in blood and urine. The exception was the last blood-urine pair in patient 3.3, sampled at 23 months post KTx. At this time point, the blood *VP1* sequence carried three new non-synonymous BC-loop mutations in addition to the four mutations that were present in the paired urine sample, or had been detected in previous urine samples.

Furthermore, *VP1* sequences were obtained from urine samples of KTx recipients in Nantes and Leipzig, and pooling these results with the longitudinal data, 71 patients (54 with genotype I including 28 full-length VP1 sequences; 17 with genotype IV virus) with one or more BC-loop mutation were identified. In this cross-sectional sample, the pattern of mutations in the BC-loop differed slightly between genotype I and genotype IV viruses (Figure 2A). The most frequently mutated amino acids were D60, A72, E73, D75, and E82 in genotype I viruses, and D62, R69, E73, and D77 in genotype IV viruses. In genotype I viruses, A72V was strongly correlated with the presence of a mutation at E73 (*p* = 0.002, Fisher’s exact test). In genotype IV viruses, mutation at E73 was associated with R69K (*p* = 0.009) as well as A72V, although in the latter case, the association did not reach statistical significance (*p* = 0.051) due to small sample size. Outside the BC-loop, the only major mutation in genotype I VP1 was observed at H139. Projecting these amino-acids onto the VP1 pentamer showed that mutations in both genotype I and genotype IV VP1 were concentrated at sites forming a ring around the sialic acid binding pocket (Figure 2B,C).

To further investigate the kinetics of BC-loop evolution in non-controller patients, we analyzed VP1 sequences in sequential urine samples from four non-controller and four controller patients by NGS. At early time points, virus populations were dominated by wild-type VP1 sequences, with <5% mutant sequences detected in all patients. BC-loop mutations then rose to frequencies >10% in non-controller patients after several months of persistent viruria and viremia (Figure 3). A recurring pattern was the emergence of one BC-loop mutation (E73Q in patient 3.5, E73A in patient 3.4, and D77N in patient 3.3) followed by the addition of further mutations to the same mutant *VP1* allele. In contrast, BC-loop mutations were not detected at greater than 5% prevalence in the rare controller patients who had a peak of intense virus replication, then subsequently had prolonged low-level viruria and viremia (patients 3.1, 3.2 and 3.12).

### 3.2. Effects of VP1 Mutations on Infectivity of Genotype I and Genotype IV PSV

To investigate the functional impact of BC-loop mutations, four different genotype Ib2 VP1 variants and two different genotype IVc2 VP1 variants observed in non-controller patients were selected for further study (Appendix A). Firstly, the combinations of different BC-loop mutations were incorporated into expression vectors coding for genotype Ib2 and IVc2 VP1 proteins, then these plasmids were used to prepare pseudotype viruses (PSV) as previously described [35]. We noted that the genotype Ib2 plasmid [37] coded for a VP1 protein carrying two mutations (E73K and E82D). These were reverted to the wild-type amino-acids before incorporating further mutations. Some mutant VP1 plasmids failed to support PSV production, or were produced at very low titres that precluded their use (Appendix A), so it was not possible to examine the biological properties of all the VP1 variants observed in vivo.

Infectivity, quantified by the ratio of infectious particles to reporter gene copies in purified PSV stocks, was measured for wild-type PSV, and PSV carrying different VP1 variants. In all cases, the VP1 variants that emerged in patients were associated with reduced infectivity in 293TT cells (Figure 4A,D). The variants Ib2-V^68^V^72^A^73^, Ib2-V^72^K^73^, and Ib2-V^72^Q^73^Q^82^ all had approximately 50-fold lower infectivity than the wild-type Ib2-E^73^E^82^ PSV in 293TT cells, while Ib2-N^60^N^69^V^72^Q^82^ resulted in only a 5-fold loss in infectivity. Mutations at the conserved E73 residue significantly reduced PSV infectivity, particularly for E73K (Figure 4B), and an incremental effect of the accumulation of multiple mutations was observed for the Ib2-V^72^Q^73^Q^82^ variant (Figure 4C).

Similar, though less severe, effects were observed in the two genotype IV variants that were analyzed. In a genotype IV context, the E73K mutation resulted in a 10-fold reduction in infectivity in 293TT cells (Figure 4D), while the IVc2-S^61^K^69^A^73^N^77^ variant had only 3 to 4-fold lower infectivity than wild-type IVc2-N^61^R^69^E^73^D^77^ or the IVc2-S^61^R^69^E^73^D^77^ single mutant (Figure 4D). The N61S mutation, which was present at the initial peak of viruria, appeared to be a neutral polymorphism, as it did not affect infectivity in 293TT cells. D77N reduced infectivity in 293TT cells, and this was mitigated by the R69K mutation. The addition of the E73A mutation slightly reduced infectivity to that seen in the gIVc2-S^61^R^69^E^73^N^77^ variant (Figure 4E).

All of the variants tested were found in *VP1* sequences from patients with high viral loads in which the mutant variant dominated the viral population and, therefore, represented viruses that replicated efficiently in vivo. To confirm that VP1 variants with strongly reduced infectivity in 293TT cells were derived from replication-competent viruses, the genotype I BKPyV-MM virus was mutated to incorporate V^72^K^73^, and the V^72^Q^73^Q^82^ VP1 mutations. Circularized genomic DNA was transfected into Vero cells, and efficient replication of both these mutant viruses was observed at 14 days post-transfection by nuclear staining for LT antigen (Figure 4F and Appendix A). Indeed, both variants reproducibly gave a higher percentage of LT antigen positive cells than wild-type plasmid in this assay, with the V^72^K^73^ variant seemingly replicating more efficiently than V^72^Q^73^Q^82^. Since VP1 variants with reduced infectivity in 293TT retained infectivity in renal epithelial cells, we then measured the infectivity of PSV incorporating different VP1 variants in the RS cell line, which are human renal tubular epithelial cells (RTEC) immortalized by the SV40 LT antigen and, therefore, a better model for primary RTEC than 293TT cells. VP1 mutations had a markedly reduced impact on PSV infectivity in RS cells (Figure 5A,B). Among genotype Ib2 variants, Ib2-N^60^N^69^V^72^Q^82^ had 3-fold lower in infectivity in RS cells compared to Ib2-E^73^E^82^, whereas Ib2-V^68^V^72^A^73^, Ib2-V^72^K^73^, and Ib2-V^72^Q^73^Q^82^ all had infectivity close to wild-type in RS cells (Figure 5A). Similarly, the IVc2-S^61^K^69^A^73^N^77^ and IVc2 K73 variants had infectivity close to wild-type IVc2. (Figure 5B). Furthermore, at PSV doses greater than 200 pEGFP copies per cell, the percentage of GFP^+^ RS cells observed after infection with Ib2-V^72^Q^73^Q^82^ and IVc2-S^61^K^69^A^73^N^77^ variant PSV was significantly higher than for the corresponding wild-type PSV (Figure 5C,D). In contrast, in 293TT cells, the percentage of GFP^+^ cells compared to wild-type was significantly lower at all tested doses (Figure 5E,F). Therefore, the Ib2-V^72^Q^73^Q^82^ and IVc2-S^61^K^69^A^73^N^77^ variants showed increased capacity to infect RS cells, but reduced infectivity in 293TT cells compared to wild-type gI and gIV PSV.

Treatment of cells with neuraminidase prior to infection with PSV fully inhibited Infectious entry of Ib2-E^73^E^82^ and Ib2-V^72^Q^73^Q^82^ PSV into both 293TT cells and RS cells (Figure 6A), indicating that the Ib2-V^72^Q^73^Q^82^ VP1 variant remained sialic acid dependent. In contrast, entry of genotype IVc2 PSV into 293TT cells was only partially inhibited by neuraminidase treatment (Figure 6B), while infection by IVc2-S^61^K^69^A^73^N^77^ PSV in RS cells was significantly more resistant to neuraminidase treatment than the corresponding wild-type PSV (Figure 6B).

### 3.3. Effects of Mutations on Neutralization

Since some VP1 variant PSV had drastically reduced infectivity in 293TT cells, it was necessary to use high doses of these PSV in neutralization assays. Control experiments showed that when used at 5 × 10^6^ and 1 × 10^6^ EGFP copies per well, the neutralizing titre against gIb2-E^73^E^82^ PSV measured in a panel of six patient sera was significantly lower at the higher PSV dose. However, when neutralizing titres against Ib2-E^73^E^82^ and gIa PSV (which carry the same BC-loop sequence) were compared at the same input dose, neutralizing titres were not significantly different (Appendix A). Neutralizing assays comparing PSV with wild-type and variant VP1 were therefore conducted at the same input dose of PSV against a panel of sera drawn at 12 months post-KTx from six controllers and six non-controllers. To compare the effects of different VP1 mutations on neutralization escape, variant-specific neutralizing titres in each serum were normalized by subtracting the log10 neutralizing titre against wild-type VP1 from the log10 neutralizing titre observed against the variant in the same serum sample. Neutralization escape, defined by a lower neutralizing titre against the variant than against the wild-type, therefore gives negative values on this scale.

Taken as a group, sera from non-controllers were not more susceptible to neutralization escape than sera from controllers (Figure 7A,B). However, in three non-controllers (3.3, 3.4 and 3.8), neutralization escape of the patient’s VP1 variant (red symbols, Figure 7A,B) by cognate serum was much clearer than in any of the 11 other sera tested against that variant. The Ib2-N^69^Q^82^ variant that emerged in patient 3.4 was 10-fold more resistant to neutralization by cognate serum than wild-type, while the Ib2-N^60^N^69^V^72^Q^82^ variant was fully resistant to neutralization (Figure 7C). Similarly, the IVc2-S^61^K^69^A^73^N77 variant from patient 3.3 was 17-fold more resistant to neutralization by cognate serum than wild-type IVc2 PSV. Similar results were obtained in RS cells (Appendix A), although due to the lower infectivity of the Ib2-N^69^Q^82^ and the Ib2-N^60^N^69^V^72^Q^82^ variants in RS cells, it was not possible to determine whether these VP1 variants were resistant to neutralization in this cell line. Further experiments confirmed that the combination R69K plus D77N found in patient 3.3 only conferred resistance to neutralization by cognate serum (Appendix A). These VP1 mutations were therefore specifically adapted to the neutralizing profile of the host’s serum. 

## 4. Discussion

In the present work, we investigated the evolution of BKPyV *VP1* in patients who showed no virological response to modulation of immunosuppressive therapy compared to controllers, whose viral load dropped more than 1000-fold in the 6 months following peak viruria. In a prospective cohort of KTx patients recruited in Nantes in 2011–2014, non-controllers made up approximately 30% of the patients (7 out of 24) in whom viruria was greater than 7 log10 copies/mL. Although the patient group definitions are different, this proportion is similar to the 26% rate of persistent PyVAN reported by the Banff working group on PyVAN [12]. Importantly, recent analysis indicates that persistent PyVAN is associated with an increased risk of graft failure and that graft loss occurs almost exclusively in patients with persistent PyVAN [38]. Understanding the physiological basis of persistent, high-level BKPyV replication in KTx recipients is therefore important in order to mitigate these adverse clinical outcomes.

In the present work, most clinical and biological parameters were similar in controllers and non-controllers, and in particular there was no difference in induction or maintenance immunosuppression between the two groups. However, CIT was two times longer in controllers compared to non-controllers, and this difference remained significant after exclusion of live donor transplants. Although this result should be interpreted with caution due to the small sample size, it suggests that when high-level BKPyV replication occurs, it may persist longer in KTx recipients whose transplant was characterized by short CIT. The biological basis of this observation is not clear, since cold ischemia time has not been reproducibly identified as a risk factor for BKPyV reactivation post-graft [39,40]. However, longer CIT is a clear risk factor for delayed graft function and acute rejection [41] and is also related to chronic graft damage [42]. If the same immunological mechanisms, such as higher levels of intra-graft inflammation and T-cell infiltration [43], are involved in graft rejection and control of BKPyV reactivation, this could lead to an association between longer CIT and more rapid control of BKPyV replication.

We found that in all KTx recipients with BKPyV reactivation, the virus population was initially dominated by wild-type *VP1* sequences and that non-synonymous *VP1* mutations then accumulate over time. The emergence of VP1 mutations was directly related to the duration of high-level BKPyV replication, with the accumulation of non-synonymous BC-loop mutations observed both by Sanger and NGS sequencing. The accumulation of VP1 mutations over time was initially reported in sequential biopsies from KTx recipients with PyVAN [22], and more recently, Peretti et al. found that VP1 mutations were present in blood and urine from two patients with PyVAN, but not in plasma or urine samples from 16 KTx recipients with ongoing BKPyV replication in the absence of PyVAN [26], suggesting that BC-loop mutations are a specific hallmark of PyVAN. Our results do not support this interpretation, since BC-loop mutations also emerged in patients with persistent high-level viruria in the absence of viremia and with normal graft function (eg. patients 2.4 and 3.9 Figure 1F), who had no suspicion of PyVAN. Furthermore, VP1 variants were only found at a prevalence of less than 5% in the viral population in patients with acute PyVAN, accompanied by transient high-level viruria and viremia (eg. patient 3.1 and patient 3.5 at initial peak viruria Figure 3). Therefore, BC-loop mutations are not found in all patients with PyVAN, nor are they restricted to patients with PyVAN. In fact, patient 3.9 illustrated both of these characteristics in the same individual (Figure 3). This patient had two consecutive kidney grafts, with the second following graft loss due to PyVAN. At the time of the first histological PyVAN diagnosis, the E73A mutation was only rarely detected (PyVAN without prevalent VP1 mutations). Following the second graft, the initial virus persisted, and accumulated several BC-loop mutations, even though graft function was normal and viremia was undetectable (VP1 mutations without signs of PyVAN). These counterexamples convince us that the duration of continuous BKPyV replication, and not the presence of PyVAN, is the key determinant for the emergence of BC-loop mutations in KTx recipients. Our data are consistent with the first longitudinal study [22] of VP1 variability, which also demonstrated the accumulation of mutations over time, and two subsequent studies that did not find a relationship between PyVAN and the presence of BC-loop mutations [23,44]. Some inconsistency between studies may be caused by patient stratification into PyVAN/viremia/viruria groups, which are then compared cross-sectionally. Most patients with viremia and/or viruria without PyVAN rapidly control virus replication, while many patients only develop PyVAN after a prolonged period of virus replication, and this may lead to some degree of confounding between cross-sectional groups and the duration of virus replication. To the best of our knowledge, the present work is the first study in which the duration of virus replication was specifically taken into account in addition to the distinction between PyVAN/viremia/viruria, and this is what allowed us to document the accumulation of VP1 mutations in patients with persistent viruria in the absence of viremia, and the absence of VP1 mutations in patients with acute PyVAN. Whether VP1 evolves more rapidly in patients with PyVAN than in patients with only persistent viruria, and whether the rate of VP1 evolution differs between genotypes are open questions that will require the study of a larger patient cohort.

With respect to the functional impact of BC-loop mutations, very few neutral mutations (only N61S and R69K in gIV) were identified. All other variants, and in particular mutations that modified the conserved E73 residue, resulted in a significant loss of infectivity in 293TT cells. Wild-type gI and gIV VP1 sequences therefore appear to occupy a fitness peak, and this could be related to the high degree of VP1 conservation that is observed in BKPyV genomes. In most cases, however, the diminished infectivity we observed in 293TT cells was not recapitulated in either Vero cells or RS cells, which is consistent with previous data on the infectivity of patient-specific VP1 variants in Vero cells [45]. Indeed, both the Ib2-V^72^Q^73^Q^82^ and IVc2-S^61^K^69^A^73^N^77^ variants showed significantly higher infectivity in RS cells than gIb2 and gIVc2 wild-type PSV, despite having significantly lower infectivity in 293TT cells. These results suggest that entry receptors on 293TT cells and RTEC may not be the same, and our results are consistent with the observation that BC loop mutants are able to engage a different spectrum of cell surface glycans than the WT pseudovirus [26]. The one exception was the Ib2-N^60^N^69^V^72^Q^82^ variant, which had a similar loss of infectivity in both 293TT and RS cells (Figure 5A). Unlike JCPyV, in which VP1 mutations have been shown to switch virus tropism to non-sialylated entry receptors [46], we found that infectious entry of PSV with the Ib2-V^72^Q^73^Q^82^ variant was still sialic acid-dependent in both 293TT and RS cells (Figure 6A). Genotype IV viruses may, however, use some non-sialylated entry receptors, since entry of PSV with both gIVc2-Wt and IVc2-S^61^K^69^A^73^N^77^ into 293TT was not fully blocked by neuraminidase treatment, and entry into RS cells mediated by IVc2-S^61^K^69^A^73^N^77^ mutant VP1 was almost entirely insensitive to neuraminidase (Figure 6B). The nature of these potential non-sialylated entry receptors, and the putative receptors for VP1 variants such as Ib2-V^72^Q^73^Q^82^, remains to be determined.

BC-loop mutations conferred neutralization escape from cognate serum in five out of six non-controller patients studied. In three of these patients (3.3, 3.4 and 3.8), the mutations that arose in vivo were specifically adapted to the host’s humoral response, strongly indicating that VP1 mutations accumulate through positive selection for neutralization escape variants in these individuals. In the two other patients (3.5, 2.4), the VP1 mutations that were observed did confer some degree of neutralization escape from cognate serum, but this was not greater than that seen with serum from other controller or non-controller patients. Only the Ib2-N^60^N^69^V^72^Q^82^ variant observed in patient 3.4 conferred complete neutralization escape, and this was also the only VP1 variant that displayed markedly reduced infectivity in RS cells, implying that the resistance of this mutant to host neutralization constituted a sufficient selective advantage to compensate its loss of infectivity in RTEC. Unfortunately, it was not possible to confirm that the Ib2-N^60^N^69^V^72^Q^82^ variant was able to infect RS cells in the presence of serum from patient 3.4, since the infectivity of this variant was too low in RS for neutralization assays to be carried out successfully. Nevertheless, since neutralization assays gave comparable results in 293TT cells and RS cells (compare Figure 7 and Appendix A) for the other VP1 variants, it is likely that that the neutralization escape observed in 293TT cells for Ib2-N^60^N^69^V^72^Q^82^ is relevant to neutralization resistance of this variant in RTEC. 

For the five other variants studied, the quantitative level of neutralization resistance was relatively modest, ranging from roughly 2 to 3-fold (0.3–0.5 log10) lower neutralizing titres for Ib2-V^72^Q^73^Q^82^, to more than 10-fold (1.1–1.2 log10) lower neutralizing titres for gIV variants. The minor impact of Ib2-V^72^Q^73^Q^82^ on neutralization by serum from patient 3.5 and the susceptibility of Ib2-V^68^V^72^A^73^ to neutralization by serum from patient 3.9 are not consistent with neutralization escape as a significant factor in VP1 evolution in these patients. However, it should be noted that in both these patients, further VP1 mutations accumulated at later time points (K69M and D77N in patient 3.5 and S77I and N62H in patient 3.9, Figure 2), and it is possible that significant neutralization escape would have been observed if it had been possible to prepare PSV, incorporating these further mutations. It is perhaps also pertinent that the serum from patient 3.9 used in neutralization assays was drawn at 23 months post-KTx, at which point Ib2-V^68^V^72^A^73^ had been the dominant variant for more than 12 months, and it is therefore possible that patient 3.9 had generated a specific response against this variant. Alternatively, since Ib2-V^72^Q^73^Q^82^ PSV showed enhanced infectivity in RS cells compared to wild type, this variant might have emerged by outcompeting wild-type virus for infection of RTEC, independently of any selection pressure exerted by the humoral response. Previously published modelling work has indicated that the majority of virus shed into the urine originates from infected cells in the urothelium rather than the tubular epithelium in the kidney [47]. If the entry receptors on urothelial cells are slightly different from those present on RTEC, this could explain the otherwise puzzling observation that the highly conserved wild-type VP1 sequences of both genotype I and genotype IV BKPyV are highly adapted to entry into 293TT cells, but do not represent the optimal solution for infecting RS cells. In this scenario, PSV entry into 293TT cells would be a model for BKPyV infection of urothelial cells, whereas PSV entry into RS cells would more closely represent infection of RTEC, and the accumulation of VP1 mutations observed in patient 3.5 could reflect a progressive specialisation of the virus towards RTEC rather than escape from neutralization. Of course, these two processes are not mutually exclusive, for example the IVc2-S^61^K^69^A^73^N^77^ variant showed both resistance to neutralization and enhanced infectivity in RS cells. 

Overall, our results confirm and extend the findings of Peretti et al. [26] and underline the role of the humoral response as a driver of VP1 evolution in patients. Neutralizing antibodies therefore can exert selection pressure on BKPyV in patients with persistent viruria, but appear not to be sufficient, on their own, to eradicate the virus. This may occur because the concentration of IgG in urine, and presumably the glomerular filtrate, of healthy adults is at least 1000-fold lower than in serum [48], so high neutralizing titres in the serum may not fully block virus transmission in the urothelium or renal tubules [49]. Of course, the humoral response is not deployed in isolation, and one important limitation of the present work is that BKPyV-specific CTL responses were not measured. However, in the light of several previous studies that have shown the importance of the antiviral CTL response for BKPyV clearance [15,16,17], it is likely that our non-controller patients did not mount an effective BKPyV-specific CTL response. This may in fact be the key event that allows the virus capsid to evolve, in the sense that if CTL fail to kill infected cells, then neutralizing antibodies become the last line of defence, and their impact on viral fitness becomes more important. If cellular and humoral responses both fail—in the latter case, due to the selection of VP1 mutants that confer viruses with a competitive advantage in terms of neutralization escape—the result is the persistent high level BKPyV replication that characterizes the non-controller group.

Clinical management of PyVAN currently relies on diminution of immunosuppressive therapy, but this strategy does not always lead to control of virus replication and resolution of nephropathy. It would therefore be useful to identify predictors of the virological and clinical response to modulation of immunosuppression. For this reason, we compared the specificity of the neutralizing response in a panel of BKPyV controllers and non-controllers, working under the hypothesis that serum from non-controllers would have a narrower neutralizing response than serum from controllers. Contrary to our hypothesis, however, no clear difference in specificity between controller and non-controller sera was observed. For example, the Ib2-V^72^Q^73^Q^82^ and IVc2-K^73^ variants were partially resistant to neutralization by sera from both controller and non-controller patients. Therefore, the neutralizing profile of sera from KTx recipients patients with BKPyV reactivation is unlikely to have clinical value as a predictor of persistent high-level virus replication. On the other hand, the existence of escape mutations has implications for therapy of BKPyV with IVIG [50] or monoclonal antibodies. For example, the D60N mutation observed in patient 3.4 disrupts the epitope bound by the broadly neutralizing monoclonal 41F17 [51] and may confer neutralization escape to this antibody. Firstly, this implies that a combination of different monoclonals recognizing distinct neutralizing epitopes may be required for optimal PyVAN therapy. Secondly, since escape mutations accumulate over time, preventive or pre-emptive use of neutralizing monoclonal antibodies may prove more effective than a curative approach.

## Figures and Tables

**Figure 1 viruses-12-00824-f001:**
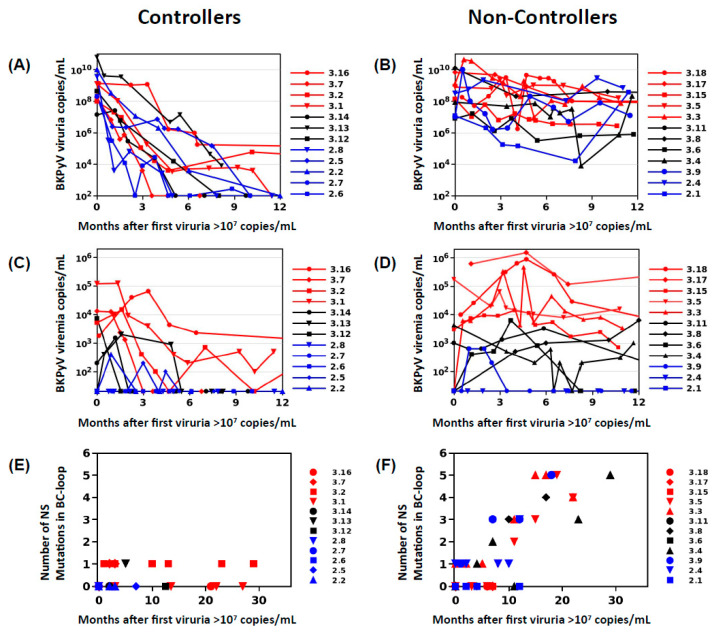
Evolution of viruria and VP1 mutations in controller and non-controller groups. Urine (**A**,**B**) and blood (**C**,**D**) viral load over time in controller (**A**,**C**) and non-controller (**B**,**D**) patients. Symbol colours correspond to high (>4log10 copies/mL, red), intermediate (between 3log10 and 4log10 copies/mL, black), and low (<3log10 copies/mL, blue) peak viremia. Accumulation of non-synonymous VP1 BC-loop mutations in non-controller (**F**), but not in controller patients (**E**). For each sample analyzed by PCR and Sanger sequencing, the number of differences compared to wild-type was plotted against time after first viruria >10^7^ copies/mL. Blue symbols in panel E show patients who accumulate BC-loop mutations in the absence of viremia.

**Figure 2 viruses-12-00824-f002:**
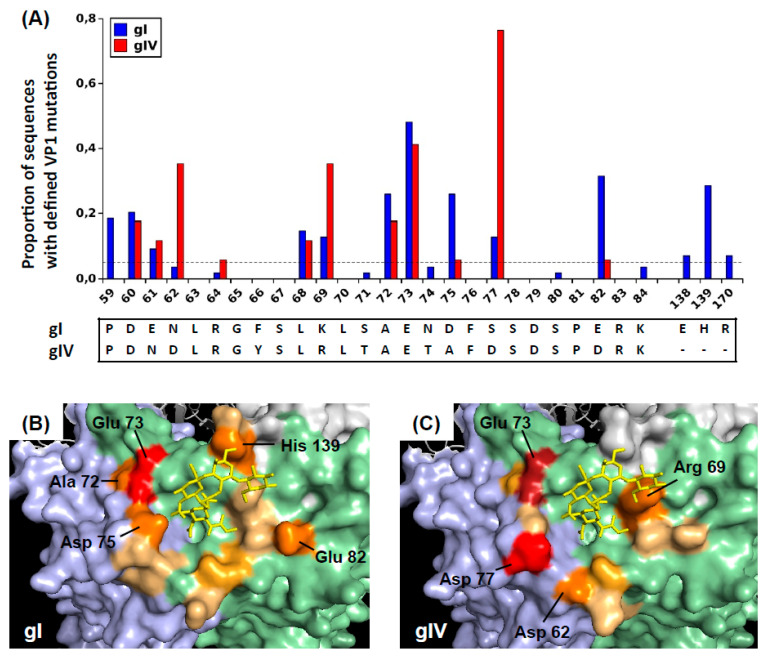
Distribution of VP1 mutations observed in patients with genotype I and genotype IV virus. (**A**) Prevalence of VP1 mutations among sequences from 54 patients with genotype I BKPyV (26 typing region sequences AA 55–110, 28 full-length VP1 sequences) and 17 patients with genotype IV virus (typing region sequences only). Localisation of observed mutations on genotype I (**B**) and genotype IV (**C**) VP1 pentamers. BKPyV VP1 pentamers (4MJ0 [36]) were visualized with Pymol, and the genotype IV-specific BC-loop residues were added using the mutate function. Adjacent VP1 monomers are coloured light blue, light green and light grey, and the GD3 oligosaccharide is shown in a yellow stick representation. Mutations with prevalence >5% are coloured in progressively darker shades on the following scale: mutation prevalence 5–15% lightorange; 15–25% brightorange; 25–35% orange; 35–45% firebrick; >45% red. The files used to prepare panels (**B**) gI_VP1Muts_Orange.pse and (**C**) gIV_VP1Muts_Orange.pse are included in the Appendix A.

**Figure 3 viruses-12-00824-f003:**
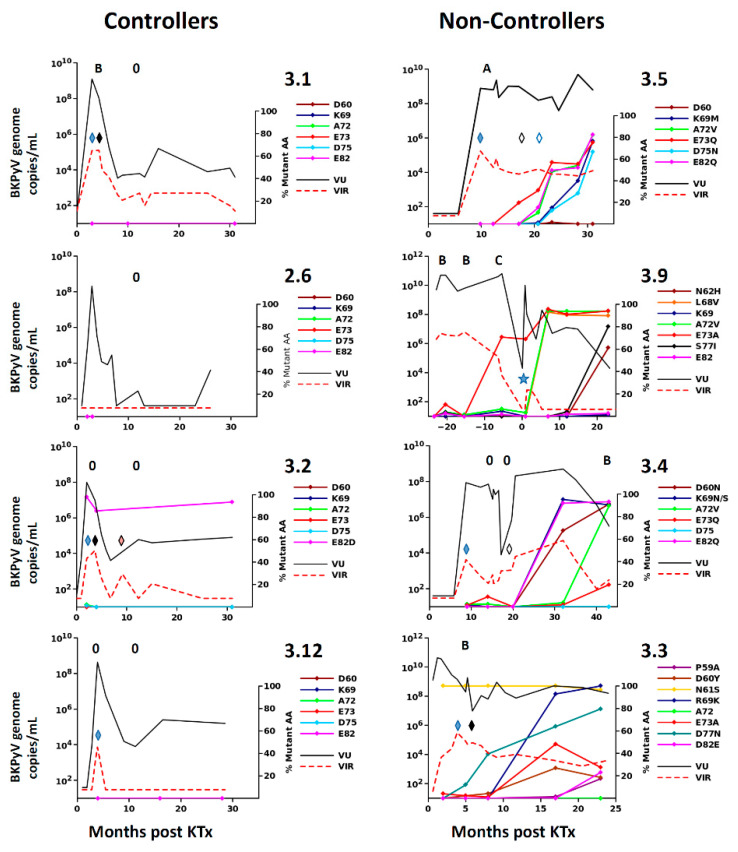
Prevalence of BC-loop mutations in virus population studied longitudinally in controllers and non-controllers. In each panel, urine viral load (VU) and viremia (VIR) are shown against time after Ktx with the scale on the left axis, together with the proportion of selected BC-loop mutations (right axis) detected in the urine at different times. Left panels: controllers; right panels: non-controllers. In non-controllers, the BC-loop mutations that made up at least 10% of the virus population in at least one sample are shown. In controllers, the most frequent BC-loop mutations observed in non-controllers are shown. Mutations at other sites were not detected in controllers. Absence (0) or presence of stage A, B, or C PyVAN in graft biopsies is indicated. Modifications of immunosuppression are indicated by lozenges. Filled blue—MMF reduced by 50%; filled black—MMF discontinued; open black—switch to MPO; open blue—MPO dose reduced by 50%; pink—MMF resumed at 50% dose. In patient 3.9, time is indicated in months post second KTx. Viral load and biopsy results before T = 0 refer to the first KTx. No biopsies were performed for the second graft, since renal function was normal and viremia was negative. The star symbol indicates the initiation of a reduced immunosuppression regime of azathioprine plus low dose MMF.

**Figure 4 viruses-12-00824-f004:**
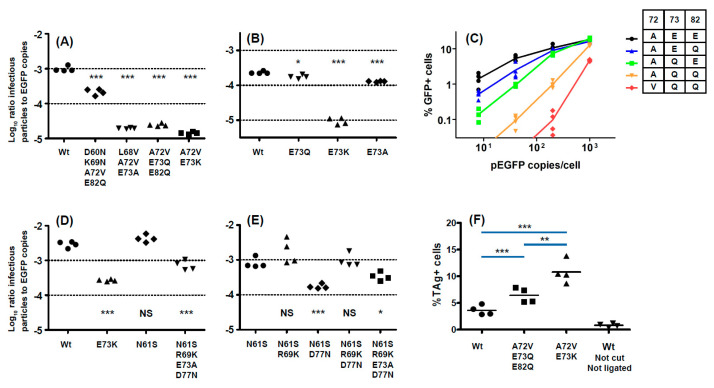
Effects of BC-loop mutations on pseudotype and virus infectivity. Infectivity in 293TT cells of PSV with wild-type genotype Ib2 VP1, and VP1 with multiple (**A**) or individual (**B**) BC-loop mutations. Cumulative effect of A72V, E73Q, and E82Q mutations on infectivity of genotype Ib2 PSV in 293TT cells (**C**). Infectivity in 293TT cells of PSV with wild-type genotype IVc2 VP1, and VP1 with multiple BC-loop mutations (**D**). Cumulative effect of R69K, E73A and D77N mutations on infectivity of genotype IVc2 PSV in 293TT cells (**E**). Replication in Vero cells of wild-type BK-MM virus, and mutant viruses incorporating A72V, E73K and A72V, E73Q, E82Q VP1 mutations (**F**). The uncut pBKV 35-1 plasmid was used as a negative control. Significant differences between groups were tested by one-way ANOVA followed by Tukey’s post-hoc test *** *p* < 0.001; ** *p* < 0.01, * *p* < 0.05. All results shown are from one representative experiment of at least two independent experiments.

**Figure 5 viruses-12-00824-f005:**
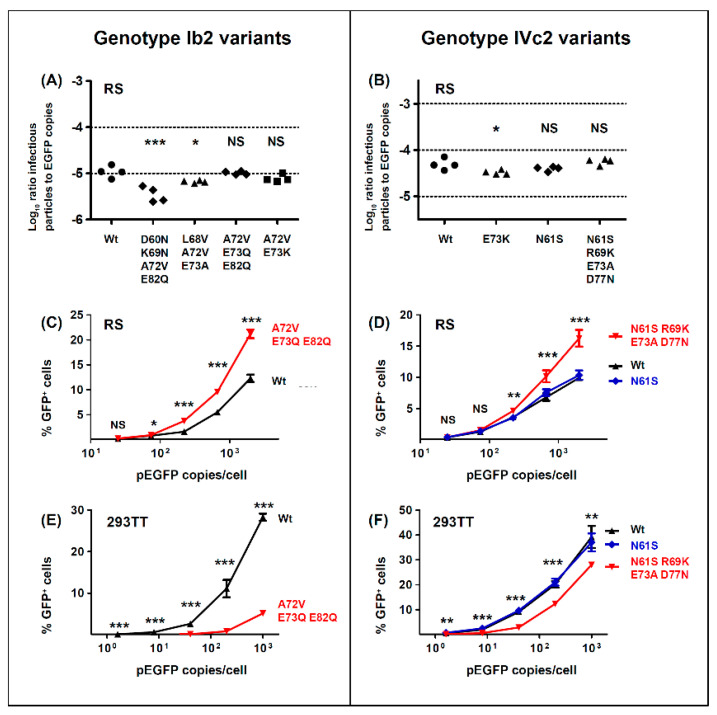
Opposite effects of BC-loop mutations on pseudo-type infectivity in RS and 293TT cells. Infectivity in RS cells of PSV with genotype Ib2 (**A**) and genotype IVc2 (**B**) VP1 variants. Percentage of infected RS cells as a function of input PSV dose for wild type gIb2 PSV compared to the V^72^Q^73^Q^82^ mutant (**C**) and wild type gIVc2 PSV compared to the S^61^ and S^61^K^69^A^73^N^77^ mutants (**D**). Percentage of infected GFP^+^ 293TT cells for the same gIb2 (**E**) and gIVc2 PSV (**F**). In the experiments shown, the gIb2 Wt and V^72^Q^73^Q^82^ PSV were produced in parallel in the same production lot, as were the gIVc2 Wt S^61^ and S^61^K^69^A^73^N^77^ PSV. The percentage of GFP^+^ cells at each PSV dose was compared by two-tailed t-test (**C**,**E**) or by ANOVA followed by Dunnett’s post-hoc test (**D**,**F**). *** *p* < 0.001; ** *p* < 0.01, * *p* < 0.05. The results of one representative experiment of at least two independent experiments are shown. Error bars indicate the StDev of quadruplicates.

**Figure 6 viruses-12-00824-f006:**
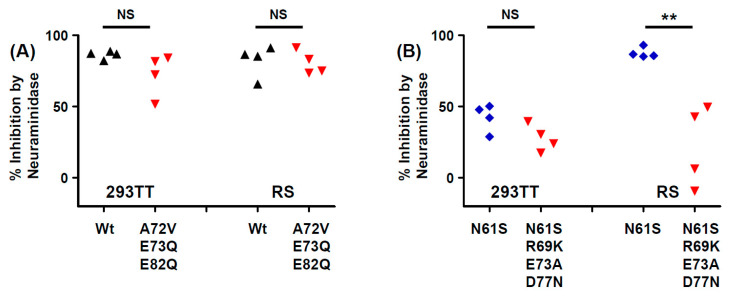
Sialic acid dependence of genotype I and genotype IV VP1 variants. Percentage inhibition of infectious entry of wild-type and V^72^Q^73^Q^82^ genotype Ib2 PSV (**A**), and S^61^ and S^61^K^69^A^73^N^77^ genotype IVc2 PSV into 293TT and RS cells (**B**). One of three independent experiments is shown, with groups compared by two-tailed t-test. ** *p* < 0.01.

**Figure 7 viruses-12-00824-f007:**
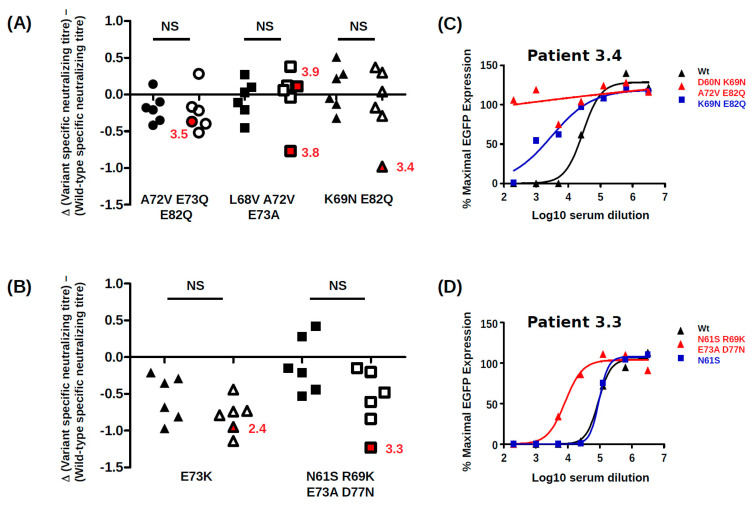
Impact of BC-loop mutations on neutralization by sera from controllers and non-controllers. Neutralizing titres in controller (*n* = 6, filled symbols) and non-controller (*n* = 6, open symbols) sera at 12 months post-KTx were measured against PSV carrying wild type VP1 and different VP1 variants. Experiments were performed in 293TT cells. For each serum, the log10 neutralizing titre against the wild-type virus was subtracted from the log10 neutralizing titre against the VP1 variant measured at the same PSV dose, and the result is plotted for each PSV variant. (**A**) genotype Ib2 variants; (**B**) genotype IVc2 variants. Red symbols and adjacent patient codes indicate the cognate non-controller serum for each VP1 variant. Input PSV doses were 500 pEGFP copies/cell for Ib2-V^72^Q^73^Q^82^ and Ib2-V^68^V^72^A^73^, 200 pEGFP copies/cell for Ib2-N^69^Q^82^ and 100 pEGFP copies/cell for genotype IV variants. (**C**) Neutralization curves for wild-type Ib2-E73E82, Ib2-N^69^Q^82^ and Ib2-N^60^N^69^V^72^Q^82^ PSV in serum from patient 3.4 at 12 months post-KTx. (**D**) Neutralization curves for wild-type IVc2, IVc2-S^61^ and IVc2-S^61^K^69^A^73^N^77^ PSV in serum from patient 3.3 at 12 months post-KTx.

**Table 1 viruses-12-00824-t001:** Clinical and biological characteristics of BKPyV Controller and Non-controller groups. For continuous variables, mean ± standard deviation are shown for each group. Means were compared by unpaired Student t-test, and categorical variables were compared by Fisher’s exact test.

	Controllers (*n* = 12)	Non-Controllers (*n* = 12)	*p*
Age (years)	56.2 ± 14.5	50.4 ± 12.8	0.308
Sex (M/F)	7/5	10/2	0.371
Kidney donor deceased/living	12/0	9/3	0.217
Cold ischemia time (hours) all donors	17.9 ± 6.6	8.7 ± 4.8	<0.001
Cold ischemia time (hours) deceased donors	17.9 ± 6.6	10.9 ± 3.2	0.008
Number of HLA mismatches	2.7 ± 1.5	4.1 ± 1.4	0.028
Induction therapy Basiliximab/ATG	9/3	7/4	0.667
Immunosuppressive regimen base MMF or MPA plus Tacrolimus	12	11	1.000
Maintenance immunosuppression included corticoids	6	8	0.680
Modification of immunosuppressive regimen following BKPyV reactivation	7	10	0.371
Received IVIG post graft	2	0	0.478
Rejection within 12 months post-graft	1	1	1.000
Months post-graft at first positive viruria	2.9 ± 2.5	3.5 ± 2.9	0.606
Peak viruria (Log10 copies/mL)	8.8 ± 1.0	8.8 ± 1.1	0.947
Viremia positive	9	10	1.000
Peak viremia (Log10 copies/mL)	3.7 ± 1.0	4.4 ± 1.2	0.238
Confirmed PyVAN	3	6	0.400
BKPyV genotype I/IV	12/0	10/2	0.478

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
