# Peer review of "Persistent BK Polyomavirus Viruria Is Associated with Accumulation of VP1 Mutations and Neutralization Escape"

_viruses, 2020, doi:10.3390/v12080824_

Round 1

Reviewer 1 Report

The authors have adressed my concerns in and those from the two other reviewer. Good job

Author Response

In response to Reviewer 3's further comments, Figure 2B and 2C have been slightly modified, and details on Figure preparation have been added to the Materials and Methods section. We hope you agree that these modifications do not alter the interpretation of the figure.

Reviewer 2 Report

all additions and comments are consistent

Author Response

(The authors gave the same response as above.)

Reviewer 3 Report

The technical details regarding the Fig 2B and Fig C

incl Pymol is completely missing and needs to be described comprehensively and in sufficient detail for readers to re-construct the results displayed.

Author Response

The following text has been added to the Materials and Methods under the VP1 sequence analysis section,

"To illustrate the positions of these mutated amino acids on the virus capsid, BKPyV VP1 pentamers were visualized using Pymol 1.8.4.0 under a Ubuntu Linux operating system. The 4MJ0 crystal structure of wild-type genotype I VP pentamer in complex with GD3 oligosaccharide [29] was loaded, then rotated in order to present the sialic-acid binding pocket with its associated oligosaccharide ligand at the centre of the figure. The surface of the central (C), clockwise (CW), and counter-clockwise (CCW) VP1 subunits were rendered using the “show surface” command, and coloured light green, light blue, and light grey, respectively. The GD3 oligosaccharide was shown in a yellow stick representation. Since no three-dimensional structure of genotype IV BKPyV VP1 has been described, for visualization purposes, the genotype IV-specific amino acids were introduced using the Wizard/Mutagenesis function. Specifically, the S71T, N74T, D75A and S77D mutations were introduced on the CW VP1 subunit, and the E61N, F66Y, K69R and E82D mutations were introduced on the C subunit. Finally, amino acid residues that were found to be mutated in patients were coloured on a scale from light orange to red, according to the prevalence of mutations found at each site. Specifically, positions 59, 60, 61, 62, 68, 69 and 82 were coloured on the C subunit, positions 72, 73, 75, 66 and 170 were coloured on the CW subunit, and positions 138 and 139 were coloured on the CCW subunit."

In addition, the following details were added to the Figure Legend to explain the orange-red colour scale,

"Mutations with prevalence >5% are coloured in progressively darker shades on the following scale: mutation prevalence 5-15% lightorange; 15-25% brightorange; 25-35% orange; 35-45% firebrick; >45% red."

in addition, we should really thank Reviewer 3 for this attention to detail because, while panels 2B and 2C were being checked to ensure that they corresponded to the step-by-step explanation of figure preparation, we noticed that the previously submitted figure contained two errors. In panel 2B, E160 had mistakenly been indicated as a mutated site in genotype I VP1, and in panel 2C, position 61 had not been mutated to the genotype IV-specific amino acid (asparagine). These two errors have been rectified in the resubmitted manuscript, along with the changes to the text explained above. We hope that Reviewer 3 agrees that this small modification to the Figure does not alter its interpretation.

Finally, the two .pse files used to prepare panels 2B and 2C have been submitted as supplementary data files.

This manuscript is a resubmission of an earlier submission. The following is a list of the peer review reports and author responses from that submission.

Round 1

Reviewer 1 Report

Dorian McIlroy et al. investigated the relationship between persistent BK Polyomavirus replication in the urine (viruria) with accumulation of non-synonymous VP1 gene mutations and neutralization escape. For this purpose, they used urine and blood samples of 10 kidney transplant recipients with persisting viruria (non-controllers) and 10 patients who suppressed viruria (controllers).

Sequencing of urine samples identified mutations within the BC-loop of Vp1 and revealed a sequential rise of mutations for non-controller patients over time.

The author find a correlation between the number of mutations in the BC-loop of Vp1 capsid protein and the duration of high-level viruria in non-controllers. In addition, using NGS, they showed that at early time points after transplantation, less than 5% of VP1 gene mutations are detected in controllers and non-controllers, whereas mutations raise to frequencies more than 10% in non-controller patients after several months of persisting viruria and DNAemia. When investigating the functional impact of those mutations with pseudotyped BKPyV particles, the variants have reduced or higher infectivity compared to wild type depending the cell types (293TT or RS).

Most frequent mutations were located around the sialic acid binding pocket, of which E73 was conserved for genotype I and IV. Pseudotyped viruses with different BC-loop mutations derived from patients were associated with reduced infectivity in HEK293TT, but not in RS cells compared to wild-type virus, respectively.

Comparing controller and non-controller groups for neutralization escape revealed no significant differences. These findings were contrary to the authors hypothesis, but provide interesting insights in BKPyV Vp1 mutant evolution, and should be better reflected in the Abstract

The Manuscript is well written and discussed.

Comments

Minor points

  • Throughout the text, old nomenclature for the major viral capsid protein; i.e., VP1, is used and should be changed into Vp1
  • Figure 7, controller and non-controller groups are not indicated in panel (A) and (B)
  • Lane 428, total of 12 sera were tested for neutralization of which 3 showed clear neutralization escape. Therefore, it should be 8 remaining sera and not 11?

Major points

  • Some BC-loop pseudotyped viruses could not be efficiently produced due to low titers. I wonder if one could generate these variants in Vero or RS cells since for these cells no reduction in infectivity was shown?
  • The authors conclude that different entry receptors might be used in HEK293TT cells. Therefore, it would be more favorable to use Vero or RS cells for neutralization assay.
  • Are the induced mutations stable? Meaning: was the virus sequenced after the very long time of 21 days of passaging in Vero cells?
  • Lane 427, the three non-controllers cannot be identified in the Figure, patients should be indicated. In Figure 7A and B, three non-controller showed neutralization escape of the cognate Vp1 variant, it is not clear which three these are since total of 6 symbols are indicated.
  • Figure 3 should show the plasma viral loads as a dotted line. The paper by Funk et al. 2008 Am J Transplant concluded that more than 80% curtailing is needed to see a relevant decline in plasma viral loads, and even higher if a decline urine viral loads is to be seen. This should be discussed also with respect to the differences between 293TT and RS cells.

The conclusions seem exaggerated when stating that

- Persistent high-level BK polyomavirus replication in KTx recipients is therefore associated with the accumulation of VP1 mutations that confer resistance to neutralization. The chicken-and -egg question: If cytotoxic T cells are not curtailing / killing infected cells, the virus may evolve further. Please address this twist of interpretation

Minor:

Pathogenic immunosuppression – not an established term

Immunosuppression may be therapeutic – Hirsch & Steiger introduced a similar classification in the 2003 review by inherited, acquired, and iatrogenic as the wanted or unwanted by-product of medical treatments. All of these may or may not cause disease – i.e. be pathogenic.

PyV is the currently recommended abbreviation of polyomavirus -  why do the authors still use PVAN and not PyVAN?

In the current update of the ASTIDCOP guidelines (Ref 5), BKPyV DNAemia contributing to the bulk of the plasma viral loads (Leuzinger et al. 2019) is used to define probable and presumptive PyVAN, which allows to trigger timely reducing immunosuppression without the need to biopsy. This may be the most relevant point in the discussion of when to best/successfully reduce immunosuppression.

The authors write that this strategy does not always lead to control of virus replication and resolution of nephropathy – what are the current success rates?

May-be the authors should use the opportunity in their introduction to reiterate the current concepts. Moreover, if plasma viral loads consist of naked DNAse sensitive BKPyV DNA (Leuzinger et al. 2019), then neutralisation becomes a difficult concept.

In a recent review by Kaur et al. 2019, the question has been raised whether or not neutralising antibodies can get into the renal tubulus, and be able to stop the cell-to-cell spread in the kidney. Please address this issue.

Reviewer 2 Report

Mc Ilroy et al. performed a study on the presence of mutations in the VP1 typing region in KTx recipients with persistent or transient high level BKPyV viruria, then analyzed the functional impact of BC-loop mutations in genotype I and genotype IV VP1 proteins on infectivity and resistance to neutralization.

The manuscript is pleasant to read with a logical sequence of results. The results seem obvious in the pathophysiology of this viral disease but needed to be demonstrated .

However the end of the manuscript on the neutralization steps is disappointing and would deserve further clarification or to repeat the experiments:

-in figure 4 it is mentioned "Results shown are from one representative experiment of at least two independent experiments"

Thus, only intra-assay variability was assessed and not inter-assay variability. For this kind of manipulation, a high variability is known. It is therefore difficult to compare and propose significant differences between different conditions. Example still in this figure 4 for the Wt in A and B we observe for two different manipulations large differences (-3 vs -3.8)

The same applies to Part C where the ratio of the differences is a function of the concentration of pEGFP.

Part F with native virus is not described in the materials and methods and would merit an addition.

Minor points:

-line 374 "these mutant virus was observed at 21 days post-transfection". This is late, isn't there a risk of reversion?

-figure 3: how were these 8 patients selected from the cohort?  Is it possible to have data on the decrease in immunosuppressive therapy for these patients?

-Line 265 to 267. Do these 3 patients have any special characteristics for mutations? Indeed, it is common to observe such characteristics in patients without any explanation.

-figure 1 why is a patient classified as controlled when the viremia persists beyond 3 log?

-figure 2 it is mentioned "28 full-length VP1 sequences" What is the frequency of these mutations in full length vs BC loop?

Reviewer 3 Report

The paper by Dorian McIlroy and colleagues describes the emergence of BC loop mutation in persistantly viruric KTx and their impact on neutralization. This is a very nice piece of well conducted translationnal research. The paper is well written but some minor modifications in figures can render it more friendly. I have not detected flaws but some points can be clarified or improved.

L77. In addition to table I, include of a flow chart as supplemental figure. Patient numbering can take into account their origin (Nantes or Leipzig)

L104. It is not clear if NGS was run on full VP1 sequence

L133. Define clearly RS cells (ATCC ?) as surrogate of RPTEC

Table I. HLA mismatch can be introduced as variable. Since some PVAN were confirmed for some patients on biopsies, APOBEC expression can be informative in connection with Peretti's paper. Acronyms have to be defined

L286. Mutations detected out of BC loop can be exposed in a supplemental figure

Fig2. Y-axis definition is not clear. Define as % ?

Fig3. Is patient 3.2 a real controller ? What is its spectrum of mutations ?

Fig4. Include former aa in the name of mutants

Fig5. Include BK genotype in C to F ti improve clarity

L430. Cognate is quite ambiguous. Does the neutralization experiments were conducted on serum drawn at the same time of mutant virus identification ? Since RS cells express SV40LT, it would have been better to use them in the neutralization assays. This has to be discussed since results obtained with this cell line are more linked with real life.